

# ATLAS results on exotic hadronic resonances

**Yue Xu⋆ on behalf of the ATLAS collaboration**

Department of Physics, Tsinghua University, Beijing 100084, China

⋆ yue.xu@cern.ch

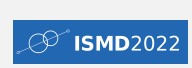

## Abstract

Recent results from the proton-proton collision data taken by the ATLAS experiment on exotic resonances are presented. A search for $J/\psi p$ resonances in $\Lambda_b \to J/\psi p K$ decays with large $pK$ invariant masses is reported. A search for exotic resonances in the four-muon final state is shown.

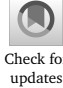
## 1 Introduction

The quark model proposed by Gell-Mann and Zweig successfully describes conventional hadrons as being composed of two or three valence quarks [1, 2]. At the same time, theorists also predicted the existence of exotic hadrons with more than three quarks, whose properties would not fit the standard picture of quark model. The study of exotic hadrons can provide a unique insight into the nature of strong interaction. This proceeding overviews the recent studies of exotic hadronic resonances with proton-proton ($pp$) collision data collected by ATLAS detector [3] at centre-of-mass energies $\sqrt{s} = 7$ [4], 8 [5] and 13 [6] TeV.

## 2 Study of $J/\psi p$ resonances in the $\Lambda_b^0 \to J/\psi p K^-$ decays

The search of potential pentaquark is carried out through $\Lambda_b^0 \to J/\psi p K^-$ decays , using 4.9 fb$^{-1}$ and 20.6 fb$^{-1}$ of $pp$ collision data at $\sqrt{s} = 7$ [4] and 8 [5] TeV. The $\Lambda_b^0 \to J/\psi p K^-$ decays are reconstructed together with $B^0 \to J/\psi K^+\pi^-$, $B^0 \to J/\psi\pi^+\pi^-$, $B_s^0 \to J/\psi K^+K^-$ and $B_s^0 \to J/\psi\pi^-\pi^+$ decays, due to lack of particle identification for charged hadrons. $J/\psi$ candidate is reconstructed through $J/\psi \to \mu^+\mu^-$ decays. Both muons are required to have $p_T > 4$ GeV and $|\eta| < 2.3$. The invariant mass of two muons is required to be $2807 < m_{\mu^+\mu^-} < 3387$ MeV. To form B-hadron ($H_b$) candidates, $J/\psi$ candidate and two additional tracks from hadrons are refitted to a common vertex with $J/\psi$ mass constraint.

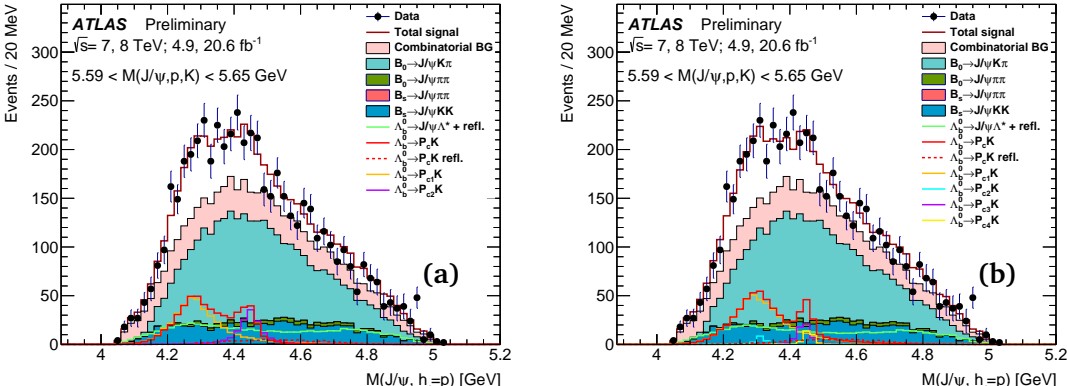

Figure 1: The $\chi^2$ fit results of the $m(J/\psi p)$ distribution in the signal region with the hypothesis of two pentaquarks (a) and with the hypothesis of four pentaquarks (b) [8].

Table 1: Extracted pentaquarks masses and widths in $\Lambda_b \rightarrow P_c^+ K^-$ decays for the hypothesis of two pentaquarks [8].

| Parameter | Value [8] | LHCb value [9] |
|---|---|---|
| $m(P_{c1})$ | $4282^{+33}_{-26}(\text{stat})^{+28}_{-7}(\text{syst})$ MeV | $4380 \pm 8 \pm 29$ MeV |
| $\Gamma(P_{c1})$ | $140^{+77}_{-50}(\text{stat})^{+41}_{-33}(\text{syst})$ MeV | $205 \pm 18 \pm 86$ MeV |
| $m(P_{c2})$ | $4449^{+20}_{-29}(\text{stat})^{+18}_{-10}(\text{syst})$ MeV | $4449.8 \pm 1.7 \pm 2.5$ MeV |
| $\Gamma(P_{c1})$ | $51^{+59}_{-48}(\text{stat})^{+14}_{-46}(\text{syst})$ MeV | $39 \pm 5 \pm 19$ MeV |

Multi-dimensional fits are performed to estimate background contributions and extract pentaquark parameters. The fit procedure is complex. It's iterative with four steps in each iteration. Parameters obtained in previous step are used in the current step. After the fit, all the pentaquark parameters can be obtained. Two hypotheses are tested in the fit:

1. Hypothesis with two pentaquarks with spin parity of $3/2^-$ and $5/2^+$, respectively.

2. Hypothesis with four pentaquarks with masses, widths and relative yields fixed to the results from LHCb [7].

Figure 1 shows the $\chi^2$ fit results of the $m(J/\psi p)$ distribution for the two hypotheses. The data description by hypotheses of two pentaquarks and four pentaquarks are both good ($\chi^2/N_{\text{dof}} = 37.1/39$ and $\chi^2/N_{\text{dof}} = 37.1/42$).

Table 1 summarizes the obtained pentaquark masses and widths for the hypothesis of two pentaquarks and compares to the LHCb values [9]. The extracted pentaquark masses and widths are consistent with the LHCb values within uncertainties.

## 3 Observation of di-charmonium events

This analysis searches for potential tetraquark (TQ) in the $4\mu$ final state through the di-$J/\psi$ and $J/\psi + \psi(2S)$ channels, using 139 fb$^{-1}$ of $pp$ data at $\sqrt{s} = 13$ TeV.

Table 2: Summary of event selection requirements for the signal and control regions [12].

| Conditions | SR | SPS CR | DPS CR |
|---|---|---|---|
| Baseline | Di-muon or tri-muon triggers, Opposite charged muons from the same $J/\psi$ or $\psi(2S)$ vertex, Loose muon ID $p_T^{\mu_1,\mu_2,\mu_3,\mu_4} > 4, 4, 3, 3$ GeV and $|\eta_{\mu_1,\mu_2,\mu_3,\mu_4}| < 2.5$ $m_{J/\psi} \in \{2.94, 3.25\}$ GeV, or $m_{\psi(2S)} \in \{3.56, 3.80\}$ GeV, Loose vertex cuts $\chi^2_{4\mu}/N < 40$ and $\chi^2_{\text{di-}\mu}/N < 100$, Vertex $\chi^2_{4\mu}/N < 3$, $L_{xy}^{4\mu} < 0.2$ mm, $|L_{xy}^{\text{di-}\mu}| < 0.3$ mm, | | |
| $m_{4\mu}$ | <7.5 GeV, | (7.5, 12.0) GeV | (14.0, 25.0) GeV |
| $\Delta R$ between charmonia | <0.25 | — | — |

Combinations of di-muon and tri-muon triggers with $J/\psi$ or $\psi(2S)$ mass window requirement are applied for the largest acceptance. Events should contain at least two opposite-charge muon pairs. The TQ vertex is reconstructed by fitting the muon tracks in the inner detector. Then, each muon pair is revertexed with a $J/\psi$ or $\psi(2S)$ mass constraint to achieve best TQ mass resolution. The TQ mass obtained with mass constraints ($m_{4\mu}^{\text{con}}$) and the one without mass constraints ($m_{4\mu}$) are both used in the analysis. The dominant backgrounds are di-charmonium production through single parton scattering (SPS) process [10] and double parton scattering (DPS) process [11], modelled by Monte Carlo (MC) simulations with normalisations and kinematics corrected in SPS and DPS control regions (CRs). The detailed event selection requirements for signal region (SR) and CRs are summarized in Table 2 [12].

Unbinned maximum likelihood fits are used to obtain TQ masses and widths in the $4\mu$ mass spectra of fit regions. Two fit regions (fit SR with $m_{4\mu}^{\text{con}} < 11$ GeV and $\Delta R < 0.25$, and fit CR with $m_{4\mu}^{\text{con}} < 11$ GeV and $\Delta R \geq 0.25$) are defined based on the baseline conditions in Table 2.

In the di-$J/\psi$ channel, three-resonance signal model with interferences among signal resonances is applied. The signal model reads

$$f_s(x) = \left| \sum_{i=0}^{2} \frac{z_i}{x^2 - m_i^2 + im_i\Gamma_i} \right|^2 \sqrt{1 - \frac{4m_{J/\psi}^2}{x^2}} \otimes R(\alpha), \tag{1}$$

where $m_i$ and $\Gamma_i$ are the masses and widths of each resonance, $z_i$'s are complex numbers representing the amplitudes ($z_1$ is fixed to unity with zero phase), the square root factor is the phase space factor, and $R(\alpha)$ is a resolution function.

In the $J/\psi + \psi(2S)$ channel, two models (model A and model B) are used, due to lower statistics. Model A considers the same resonances in the di-$J/\psi$ channel which can also contribute to $J/\psi + \psi(2S)$, plus a standalone resonance. It reads

$$f_s(x) = \left( \left| \sum_{i=0}^{2} \frac{z_i}{x^2 - m_i^2 + im_i\Gamma_i} \right|^2 + \left| \frac{z_3}{x^2 - m_3^2 + im_3\Gamma_3} \right|^2 \right) \sqrt{1 - \left( \frac{m_{J/\psi} + m_{\psi(2S)}}{x} \right)^2} \otimes R(\alpha), \tag{2}$$

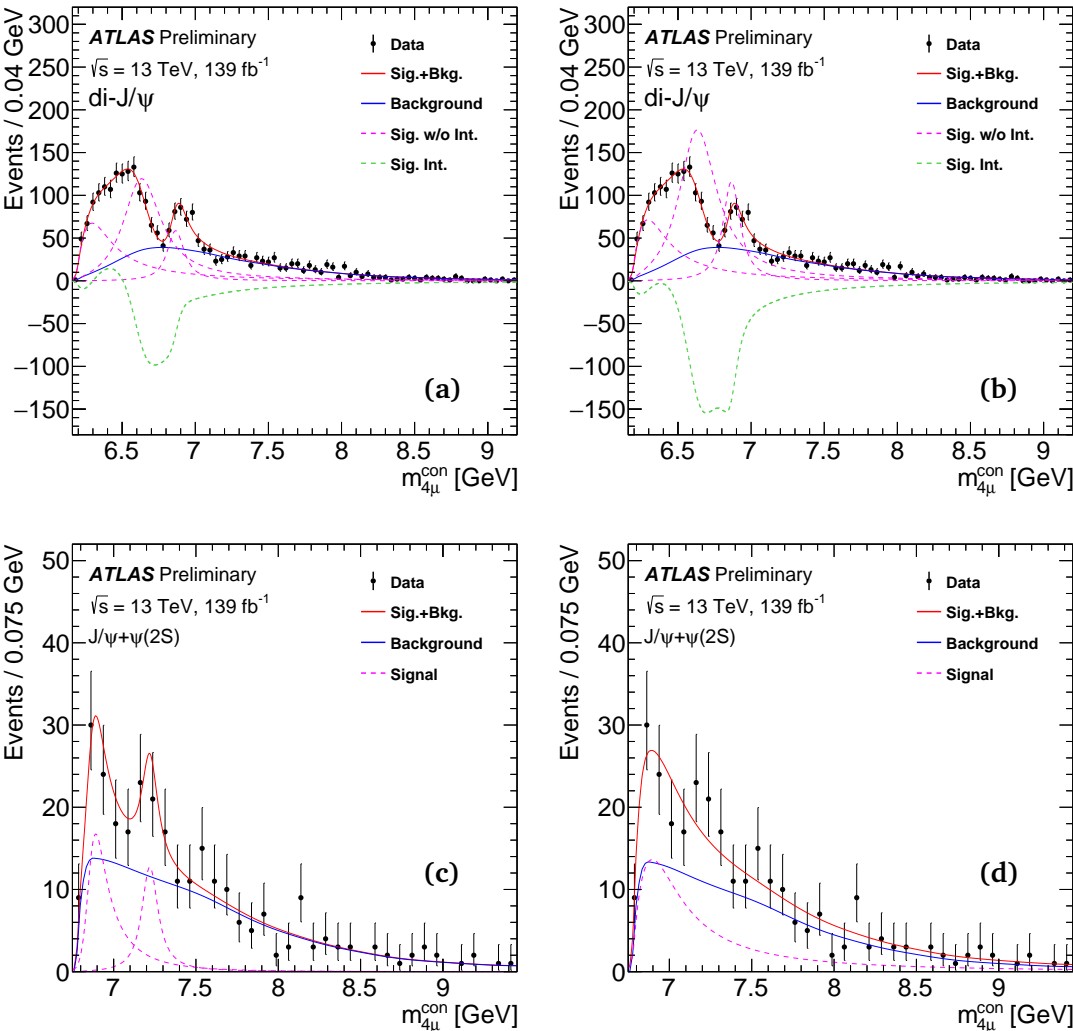

Figure 2: The fitted $4\mu$ mass spectra in the fit signal regions of the di-$J/\psi$ (a,b) and $J/\psi+\psi(2S)$ (c,d) channels. The purple (green) dashed lines represent the components of individual resonances (interferences among them). Fits in (a) and (b) have the same likelihood but different resonances magnitudes and interferences due to degenerate solutions. In the $J/\psi+\psi(2S)$ channel, both model A (c) and B (d) fit results are shown [12].

where the first three resonances are fixed to the results obtained in the di-$J/\psi$ channel. Model B only assumes a standalone resonance.

Figure 2 shows the fitted $4\mu$ mass spectra in the di-$J/\psi$ and $J/\psi+\psi(2S)$ channels. The obtained masses and widths of resonances are summarized in Table 3. In the di-$J/\psi$ channel, a broad structure at threshold and a resonance at 6.9 GeV which is consistent with the $X(6900)$ observed by LHCb [13] are observed with a significance far above $5\sigma$. Although the near threshold structure is explained by two interfering resonances, it could result from other physical processes e.g. feed down from di-charmonium resonances with higher masses.

In the $J/\psi+\psi(2S)$ channel, model A describes data better than model B, with a combined significance of $4.6\sigma$. The second resonance around 7.2 GeV with significance of $3.2\sigma$ is also hinted by LHCb [13].

Table 3: The extracted masses and widths of the resonances in the di-$J/\psi$ and $J/\psi+\psi(2S)$ channels. The first errors in each value are statistical while the second ones are systematic [12].

| (GeV) | $m_0$ | $\Gamma_0$ | $m_1$ | $\Gamma_1$ |
|---|---|---|---|---|
| di-$J/\psi$ | $6.22 \pm 0.05^{+0.04}_{-0.05}$ | $0.31 \pm 0.12^{+0.07}_{-0.08}$ | $6.62 \pm 0.03^{+0.02}_{-0.01}$ | $0.31 \pm 0.09^{+0.06}_{-0.11}$ |
| | $m_2$ | $\Gamma_2$ | — | |
| | $6.87 \pm 0.03^{+0.06}_{-0.01}$ | $0.12 \pm 0.04^{+0.03}_{-0.01}$ | — | |

| (GeV) | | $m_3$ | $\Gamma_3$ | |
|---|---|---|---|---|
| $J/\psi+\psi(2S)$ | model A | $7.22 \pm 0.03^{+0.02}_{-0.03}$ | $0.10^{+0.13+0.06}_{-0.07-0.05}$ | — |
| | model B | $6.78 \pm 0.36^{+0.35}_{-0.54}$ | $0.39 \pm 0.11^{+0.11}_{-0.07}$ | — |

## 4 Conclusion

The recent ATLAS results on exotic hadronic resonances, including pentaquark and tetraquark searches, are highlighted in this proceeding. The extracted pentaquark masses and widths are consistent with LHCb values. The potential tetraquark $X(6900)$ is observed with ATLAS data collected during Run 2. More results of exotic hadrons at ATLAS are expected in the near future with Run 2 and Run 3 data.

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
