# Peer review of "ATLAS results on exotic hadronic resonances"

_SciPost Physics Proceedings, doi:SciPost Phys. Proc. 15, 013 (2024)_

## Round 2 · Referee Report · Anonymous · 2022-11-19

Report

This manuscript provides a reasonable account of a talk given at ISMD 2022.

Requested changes

1 - The abstract states that "Studies of Zc states ... are also discussed", but these studies do not actually appear to be discussed in the manuscript. This statement needs to be corrected/clarified/removed.
2 - There is a small issue with grammar in the final sentence of the abstract. This should be changed to something like "A search for exotic resonances ..."
3 - The start of the introduction states that "The quark model of conventional hadrons successfully describes mesons and baryons as being composed of two or three valence quarks". This appears to imply that the quark model successfully describes all mesons and baryons, but this is not the case and needs to be clarified.
4 - If more details of these studies are provided elsewhere, this should be made clearer with appropriate references.
5 - In the caption of Figure 1, "SR" should be defined. (Signal region?) There also appears to be a typo in the caption: should the first "(right)" be "(left)"?
6 - In the caption of Table 1, I think "masses and widths of the" should be "masses and widths in".
7 - In the caption of Table 3 what does "natural widths" mean as opposed to just "widths"?
8 - The plots need to be labelled (a), (b), etc and the difference between (a) and (b) explained.
9 - I recommend proof reading the manuscript again.

  • validity: -
  • significance: -
  • originality: -
  • clarity: -
  • formatting: -
  • grammar: -

Author:  Yue Xu  on 2022-11-30  [id 3089]

(in reply to Report 1 on 2022-11-19)
Category:
answer to question

The proceeding has been updated. 1-4: the sentences have been corrected. 5: yes, "SR" means "signal region". The caption has been updated. 6: the caption has been updated. 7: the "natural" is removed. 8: labels are added in the figures.

---

## Editorial Decision

published